# Viral Diversity in Samples of Freshwater Gastropods *Benedictia baicalensis* (Caenogastropoda: Benedictiidae) Revealed by Total RNA-Sequencing

**DOI:** 10.3390/ijms242317022

**Published:** 2023-11-30

**Authors:** Tatyana V. Butina, Tamara I. Zemskaya, Artem N. Bondaryuk, Ivan S. Petrushin, Igor V. Khanaev, Ivan A. Nebesnykh, Yurij S. Bukin

**Affiliations:** Limnological Institute Siberian Branch of the Russian Academy of Sciences, 664033 Irkutsk, Russia; tvbutina@mail.ru (T.V.B.); ui.artem.ui@gmail.com (A.N.B.); ivan.kiel@gmail.com (I.S.P.); igkhan@lin.irk.ru (I.V.K.); canis-87@mail.ru (I.A.N.); bukinyura@mail.ru (Y.S.B.)

**Keywords:** viruses, viral diversity, RNA-containing viruses, metatranscriptomic analysis, invertebrate, gastropods, freshwater ecosystems, Lake Baikal

## Abstract

Previously, the main studies were focused on viruses that cause disease in commercial and farmed shellfish and cause damage to food enterprises (for example, *Ostreavirusostreidmalaco1*, *Aurivirus haliotidmalaco1* and *Aquabirnavirus tellinae*). Advances in high-throughput sequencing technologies have extended the studies to natural populations of mollusks (and other invertebrates) as unexplored niches of viral diversity and possible sources of emerging diseases. These studies have revealed a huge diversity of mostly previously unknown viruses and filled gaps in the evolutionary history of viruses. In the present study, we estimated the viral diversity in samples of the Baikal endemic gastropod *Benedictia baicalensis* using metatranscriptomic analysis (total RNA-sequencing); we were able to identify a wide variety of RNA-containing viruses in four samples (pools) of mollusks collected at three stations of Lake Baikal. Most of the identified viral genomes (scaffolds) had only distant similarities to known viruses or (in most cases) to metagenome-assembled viral genomes from various natural samples (mollusks, crustaceans, insects and others) mainly from freshwater ecosystems. We were able to identify viruses similar to those previously identified in mollusks (in particular to the picornaviruses *Biomphalaria virus 1* and *Biomphalaria virus 3* from the freshwater gastropods); it is possible that picorna-like viruses (as well as a number of other identified viruses) are pathogenic for Baikal gastropods. Our results also suggested that Baikal mollusks, like other species, may bioaccumulate or serve as a reservoir for numerous viruses that infect a variety of organisms (including vertebrates).

## 1. Introduction

The phylum Mollusca is the second most diverse invertebrate phylum after the Arthropoda, with approximately 85,000 described species. They are grouped into eight classes (Gastropoda, Bivalvia, Scaphopoda, Cephalopoda, Polyplacophora, Monoplacophora, Caudofoveata and Solenogastres) and inhabit a wide range of ecological niches—marine (52,525 described species), freshwater (7000) and terrestrial (24,000) habitats [1,2]. Mollusks are highly resistant to infectious diseases, do not have a classical acquired immune system, but have highly effective innate immunity [3]; as a result, the life expectancy of mollusks is from 10 to 100 and sometimes up to 400 years [4]. The resistance of mollusks to infectious diseases especially helps them survive in aquatic ecosystems, where the abundance and diversity of viruses reach huge values (up to 10^9^ or more particles per milliliter [5]). The strong immunity of mollusks not only protects against viral infections; thanks to it, these animals can be carriers of a wide range of viruses of other organisms, including humans [6,7].

The first evidence of viruses in mollusks appeared in the early 1970s. At that time, the study of mollusk viruses was carried out on the basis of histological and cytological observations using light and electron microscopy. Farley C.A., author of the review [8], summarized the known cases of virus-like particle detection in the tissues of various (more than 13 species) mollusks, mainly representatives of Bivalvia, and, based on ultrastructural morphology, systematized the putative viruses in accordance with the existing classification of viruses. Virus-like particles have been assigned to the following families (adjusted for later classification): *Herpesviridae*, *Iridoviridae*, *Papillomaviridae*, *Polyomaviridae* (DNA viruses), *Togaviridae*, *Retroviridae*, *Paramyxoviridae* and *Reoviridae* (RNA viruses). Later this list was supplemented by the families *Birnaviridae*, *Iridiviridae* and *Picornaviridae* [9,10]. With the introduction of new molecular technologies (“omics” studies), the range of viruses found in mollusks has expanded to representatives of *Totiviridae* [11], *Rhabdoviridae* [12], different families of *Picornavirales*, etc. [11,12,13,14,15,16,17,18]. To date, of the variety of revealed viruses, the best studied (including genomic characteristics) are only representatives of the families *Herpesviridae*, *Iridoviridae* and *Birnaviridae*, which cause great damage to the production of bivalve mollusks (such as oysters *Crassostrea gigas*, abalones *Haliotis diversicolor supertexta*, or scallops *Chlamys farreri*). Studies of highly pathogenic viruses of these families (especially the herpesviruses *Ostreavirusostreidmalaco1* and *Aurivirus haliotidmalaco1*) made it possible to reveal the characteristics of these viruses, the mechanisms of virus—host interaction, and also to develop measures for diagnosis, prevention and reduction in morbidity [3,19,20,21,22,23].

Currently, due to the problem of the emergence of new viral diseases and the expansion of the capabilities of molecular virology, large-scale projects are being implemented for mass sequencing (virome, metagenome or metatranscriptome studies) of various species of invertebrate, including mollusks, in nature. Thus, in the study [14], the (meta)transcriptomes of more than 220 species of invertebrates (including more than 20 species of mollusks) were studied, which resulted in the identification of 1445 RNA viruses, filling the main gaps in the phylogeny of RNA viruses and revealing the history of viral evolution. These new data also revealed frequent recombination, lateral gene transfer and complex genomic rearrangements in viruses and their hosts that contribute to their coevolution and host switching [14]. In a recently published paper [24], the authors characterized the viromes of 58 species of marine invertebrates (arthropods and mollusks) and also demonstrated the huge diversity of marine invertebrate RNA viruses; 315 new RNA viruses of various viral families and orders, including viruses from mollusks, were identified.

Metagenomic analysis, including shotgun sequencing, is also employed to identify possible viral infections during mollusk diseases. Thus, during the mass mortality events of unknown etiology among freshwater mussels (Unionida) in the rivers of the USA, several dozen new viruses were identified [15,25,26,27]. It was shown that some of them, for example, a novel densovirus (*Clinch densovirus 1*; family *Parvoviridae*), were epidemiologically linked to morbidity [25]. A number of other viruses were significantly associated with mussel mortality events, i.e., most of them were found in moribund mussels rather than apparently healthy mussels [25,26]. The authors suggested that viruses may be a direct cause of unionid morbidity or act indirectly in combination with unfavorable factors (during stress).

Most studies have primarily focused on marine mollusks, mostly bivalves and less often gastropods. Much less attention has been paid to freshwater mollusks. In our study, we used metatranscriptomic analysis to identify viruses circulating among the Baikal endemic gastropods *Benedictia baicalensis* (Caenogastropoda, Benedictiidae). The chosen approach makes it possible to identify the genomes of RNA viruses, as well as actively transcribed genes of DNA viruses, in the analyzed samples [28]. Sampling locations are indicated in Table 1 and Figure 1. More than 180 species of mollusks live in Lake Baikal; gastropods (Gastropoda) make up the largest part in terms of the number of species (150 species); they also dominate in biomass among the inhabitants of the Baikal benthic community. Baikal gastropods serve as indicators of the habitat, participate in the self-purification of the lake and play a significant role in the biological processes of the lake (mainly by destroying and recycling large volumes of organic biomass). *Benedictia baicalensis* is one of the largest, most widespread and numerous Baikal endemic species. Gastropods *B. baicalensis* live in all areas of Lake Baikal, in a wide range of depths (from 1.5 to 100 m) and biotopes, and on different substrates (rocks, pebbles and sand) [29,30]. In addition, they are omnivores (they may be scavengers, phytophagous or detritophagous in different seasons) [31]. Our study revealed a high diversity of viruses of different families and orders (mainly positive-strand RNA viruses, +ssRNA viruses) in *B. baicalensis* samples, most of which were not closely related to the known genomes of viruses. Distant similarities (most often less than 50%, in rare cases more than 70%) were found with viruses that infected or were isolated from mollusks, insects, crustaceans and other aquatic invertebrates. A tendency has been noted for a closer relationship between Baikal viruses and those from other freshwater ecosystems. This indicates the influence of habitat on the formation of viral diversity and the need to continue research into little-studied viruses of freshwater mollusks (as well as other freshwater invertebrates).

## 2. Results

### 2.1. Taxonomic Viral Composition Based on Direct Analysis of Reads

The total number of pair reads in Baikal samples (of gastropods *B. baicalensis*) were from 63 to 67 million. First, we divided the reads into major taxonomic groups and determined the taxonomic affiliation of the viral reads using the free software Kaiju v.1.9.0 (https://github.com/bioinformatics-centre/kaiju (accessed on 25 August 2023)) [32]. Kaiju implements an algorithm based on the alignment of short reads, translated into proteins, with the NCBI nr protein database. Using a similar approach, we also analyzed other transcriptomic, metatranscriptomic and viromic datasets (from studies [14,33] and some from the NCBI SRA database, Appendix A) and conducted a comparative analysis of viral sequences from different species of mollusks.

As a result, 96.3–97.1% of Baikal reads were unknown (unclassified); the rest belonged to bacterial, eukaryotic, archaeal and viral sequences (1.3–1.7%, 1.3–2.0%, 0.003–0.007% and about 0.02%, respectively) (Appendix A). The virus-like reads were assigned to the different families of DNA or RNA viruses. However, further comparative analysis was carried out only on the basis of reads similar to RNA viruses (Figure 2). As further analysis showed (based on the assembled scaffolds, Section 2.2), the overwhelming majority of sequences classified as DNA viruses turned out to be false positives.

Samples of *B. baicalensis* were distant from the main part of the datasets and were grouped together with samples of the marine gastropods *Nucella lapillus* from the UK (MolNucA.M and MolNucB.M) (Figure 2a). This similarity was determined by the large number of sequences affiliated to members of the families *Flaviviridae*, *Tombusviridae*, *Caulimoviridae*, *Hepeviridae* and *Retroviridae*; in general, the similarity of virus-like reads in samples of Baikal and marine gastropods is still difficult to explain. The most distinctive sample was MollanWu.M, which differed from the others in its habitat (Chinese land snails *Mastigeulota kiangsinensis*); the vectors indicated a significant proportion of viral sequences from the families *Picobirnaviridae*, *Bromoviridae* and *Closteroviridae* in this dataset. A certain similarity between the MollanWu.M and *B. baicalensis* samples were determined by the abundance of sequences close to the family *Partitiviridae*. Among other samples, the dataset from marine scallops *Agropecten irradians* (MolArg.M) turned out to be closer to the Baikal sets (Figure 2a). According to the comparative analysis of virotypes (Figure 2b), this sample (MolArg.M), together with MolNucA.M and MolNucB.M, clustered together with the Baikal sets. Figure 2b shows that the composition of dominant virotypes for all reference samples varied greatly.

In general, our analysis did not reveal any dependence of the distribution of samples on the taxonomic affiliation of the host (at the class level), habitat (freshwater or marine) or geographical location. It should be noted that we further detected some false positive (non-viral) reads in this analysis that belonged to host genomes (mollusks). After assembling the Baikal reads (Section 2.2), in some non-viral scaffolds (affiliated to mollusk genomes) we found small fragments encoding about 70 amino acids, 100% similar to flavivirus (shown in Figure 2b) proteins. Thus, the analysis of the “omic” datasets (including total RNA sequencing) based on short reads demonstrated the similarities not only of true viral sequences in the mollusk holobiont but also of virus-similar sequences in the genomes of the hosts. A more accurate and detailed analysis of the *B. baicalensis* viruses was carried out based on the assembled sequences.

### 2.2. Analysis of Assembled Scaffolds in Samples of Benedictia baicalensis

In total, using MEGAHIT (a specialized program for assembling short reads of metagenomic data) [34] in the mixed metagenomic assembly mode (cross-assembly), we obtained 291,903 scaffolds (≥500 nt) from which VirSorter2 (a special program for identifying viral scaffolds in metagenomic assemblies using several complementary approaches) [35] identified 611 viral scaffolds ranging in length from 500 to 11,797 bp. Based on the similarity of 1489 identified reading frames (ORFs) to known viral proteins from the NCBI database of complete viral proteomes, taxonomic affiliation and putative host were determined for 351 ORFs and finally for 283 scaffolds. Only scaffolds whose sequences belonged to eukaryotic viruses (157 viral scaffolds) were retained for further analysis in order to search for potential shellfish viruses. After additional data filtering (to remove false-positive viral scaffolds), by matching scaffolds to known gastropod genomes from the NCBI Genome database (Appendix A) and searching for transposons using the DfamScan program [36] (Section 4.6), 64 viral scaffolds remained; of these, only one was identified as a DNA virus (ssDNA, unclassified *Lake Sarah-associated circular molecule 10*), and 63 (from 24 to 33 per sample) were identified as RNA viruses (Appendix A). Thus, about half of the sequences identified by VirSorter2 as viral (mainly DNA viruses) were excluded from further analysis.

Our study of freshwater mollusks revealed a diversity of mainly novel viral genomes (assigned to virotypes of different families or unclassified virotypes). The 44 identified RNA virotypes (closely related viruses from the NCBI RefSeq database) belonged to the families *Tombusviridae*, *Nodaviridae*, *Narnaviridae*, *Qinviridae*, *Partitiviridae*, *Caulimoviridae*, *Dicistroviridae* and *Picornaviridae*. Some of the virotypes were unclassified *Picornavirales*. A larger number of RNA virotypes (29 in total) were found to be unclassified below the realm *Riboviria*; these were metagenome-assembled genomes (MAGs) from environmental samples, that is, newly discovered and poorly understood viruses. The similarity of scaffold ORFs to proteins of the identified RNA virotypes (from the RefSeq database) was 23.7–99.3%.

The heat maps (Figure 3) show all the RNA virotypes that were detected in our analysis. Samples from Listvennichny Bay collected in spring and autumn were grouped together (Figure 3), and the similarity of these samples is not surprising. Seven common virotypes were revealed between these samples (Appendix A). A separate cluster, but with longer branches, was formed by samples from Bolshiye Koty and Ushkany Islands (the number of common virotypes = 6); in this case, the similarity of samples from very distant areas is difficult to explain.

A Venn diagram (Appendix A) showed that the 11 identified virotypes were common to all samples, and several virotypes, from one to five, were found only in individual samples. In general, our data demonstrate the high diversity of RNA viruses in samples of Baikal gastropods and the circulation of similar viruses throughout the lake.

The longest RNA scaffolds (>2000 nt) are presented in Table 2 and Figure 4. The largest number of reads (~97% TPM) was accounted for by the BM_12038 scaffold with a length of 3224 bp (similar to *Changjiang tombus-like virus 1*). BM_7872 (3857 nt, virotype *Sanxia water strider virus 14*) also turned out to be quite numerous. The highest similarity of proteins (>70%) was found to proteins of the virotypes *Tiger puffer nervous necrosis virus* (99.3%), *Changjiang tombus-like virus 1* (76.4%), *Changjiang tombus-like virus 2* (76.8%), *Sanxia tombus-like virus 3* (70.8%) and *Sclerophthora macrospora virus A* (70.1%); the majority of proteins had less than 50% similarity to known viruses from the NCBI RefSeq database (Appendix A). When compared to the NCBI nr database, the similarity of proteins in some cases was somewhat greater, but overall it also remained below 50%. The longest scaffolds BM_458 (8746 nt) and BM_632 (8432 nt) had the highest similarity to the picornaviruses *Biomphalaria virus 1* and *Biomphalaria virus 3*, whose genomes were assembled from the freshwater gastropods *Biomphalaria glabrata* and *Biomphalaria pfeifferi* [11]. When compared to the NCBI nr database, scaffold BM_632 was also similar to *Bulinus globosus virus 2* (MAG from freshwater gastropods) and *scractlig virus 1* (MAG from freshwater bivalves). The list of identified virotypes also included a number of other unclassified viruses from mollusks: *Beihai picorna-like virus 15* (from octopods), *Hubei unio douglasiae virus 1* (bivalves), *Beihai partiti-like virus 2* (octopods), *Wenzhou tombus-like viruses 5* and *15* (gastropods), *Wenzhou picorna-like viruses 2* (bivalves) and *19* (gastropods) (Appendix A). The similarity with proteins of these viruses was only 24.8–49.8%. The hosts (or sources of isolation) of virotypes from the RefSeq database, in addition to mollusks, included insects, plants, fungi and vertebrates (birds, fish and carnivores) (Table 2; Appendix A).

### 2.3. Phylogenetic Analysis of RNA-Dependent RNA Polymerase Genes

Functional analysis of ORFs in scaffolds using the Pfam and CDD (Conserved Domain Database) databases revealed a number of domains characteristic of RNA viruses, for example, non-structural RNA-dependent RNA polymerases (RdRp), RNA helicase or structural Rhv and Viral_coat domains. Figure 4 shows the identified domains in the structure of the longest RNA scaffolds from *B. baicalensis* samples and their closest relatives from the NCBI RefSeq database (virotypes). The largest number of scaffolds contained *RdRp* genes, responsible for replicating the genome and transcription of RNA viruses. As known, the *RdRp* gene is a unique genetic marker for almost all RNA-containing viruses (except retroviruses), allowing us to reveal and characterize RNA viruses from “omic” data [37]. We performed a phylogenetic analysis of RNA viruses from the *B. baicalensis* samples based on the identified *RdRp* genes (Figure 5).

RdRp sequences of Baikal gastropods were distributed into five clusters corresponding to the RNA viral orders *Picornavirales*, *Tolivirales*, *Nodamuvirales*, *Durnavirales* and *Sobelivirales* (Figure 5a–c), demonstrating the distant relationship of the identified viral genomes (genomic fragments).

The largest cluster of *Picornavirales* (families *Dicistroviridae*, *Marnaviridae*, and *Picornaviridae*) included 17 RdRps of Baikal viruses (Figure 5a). The different picorna-like MAGs, isolated from the aquatic (mainly freshwater) environment or from freshwater bivalve mollusks (*scractlig virus 1*), turned out to be closest to the Baikal scaffolds. The extended lengths of the branches indicate the remoteness of the Baikal viruses from known MAGs and, to a greater extent, from characterized viruses (classified by the International Committee on Taxonomy of Viruses, ICTV). From the ICTV viruses (Figure 5a), we included the most similar viruses from different hosts (algae, pigs and goats) in the tree. The list of other less closely related ICTV picornaviruses included a variety of vertebrate viruses (of small and large livestock, fish, humans, etc.).

Tombusviruses (*Tombusviridae*) also formed a large cluster and included 18 Baikal sequences; the closest to them were MAGs from freshwater invertebrates and environment (Figure 5b). *Tombusviridae* are known to be viruses that infect plants. Moreover, tombus-like viruses were regularly found in metagenomic studies of aquatic biotopes [38], [39,40] and various invertebrate species [14]. True viruses of insects, mollusks or other organisms distant from plants are still unknown, just as it is not known whether tombusviruses can be bioaccumulated or they are mechanically transmitted by invertebrates. However, a recent analysis of insect transcriptomes (1243 species) identified insect-associated tombus-like viruses forming new clades related to the plant virus family *Tombusviridae*, which revealed the direct involvement of insects in the evolution of tombusviruses [41,42]. The small clade of nodaviruses (*Nodaviridae*), sister to tombusviruses [14,42], included three RdRp from Baikal gastropods, MAGs and viruses from marine bivalves and other invertebrates, from fish and other vertebrates (Figure 5c). It is assumed that tombus-noda viruses, together with picorna-like viruses, are the most ancient (+)ssRNA viruses of eukaryotes; moreover, plant *Tombusviridae* most likely originated from invertebrate viruses [43].

The cluster of partitiviruses (*Partitiviridae*) consisted of two subclusters and included five RdRp from *B. baicalensis* (Figure 5d). The family *Partitiviridae* (with bisegmented dsRNA genomes), includes viruses of plant, plant-parasitic fungi and protozoa (parasites that infect a wide range of mammals, birds and reptiles) [44]. Recent metatranscriptomic studies have expanded the diversity of marine partitiviruses; ‘the partiti-picobirna clade’ of dsRNA viruses [14] includes viruses infecting or associated with a broad range of divergent hosts: fungi, plants, algae, invertebrates (mainly insect and crustacean), vertebrates and protozoa. In addition to Baikal mollusks, the phylogenetic lineage of partiti-like viruses in our analysis included RdRps from a variety of terrestrial and aquatic organisms (arachnids, insects, plant-parasitic fungi, reptiles, etc.). One subcluster, which also included a scaffold from *B. baicalensis* (BM_60513), contained only genes from marine invertebrates (mainly crustaceans). It is most likely that the partiti-like sequences we discovered may have originated from viruses of fungi or protists that parasitize mollusks or other aquatic (crustaceans, algae, etc.) or semi-aquatic (water striders, waterfowl, etc.) inhabitants.

A separate lineage of sobemo-like viruses included one Baikal sequence (Figure 5e); none of the known viruses (classified by ICTV) was included in this cluster with established similarity parameters (Section 4.9). The cluster of sobemo-like viruses included viruses (MAGs) from marine crustaceans and Chelicerata, but the virus closest to the Baikal virus was *Beihai sobemo-like virus 18* from freshwater mollusks. The hosts of sobemoviruses were originally thought to be plants; however, similar to tombus-like viruses, they have also been repeatedly found in mollusks and arthropods [17,24].

Despite the discovered relationships, our data demonstrated significant distances between the RdRps of Baikal viruses and those from other organisms and biotopes (especially with known viruses from ICTV), indicating the novelty of the viruses we discovered.

## 3. Discussion

In this study we first estimated the diversity of viruses in the Baikal endemic gastropods *B. baicalensis* from three different regions of the lake by shotgun sequencing of total RNA isolated from the samples. Our data revealed the novel RNA viruses similar to those from different viral orders (*Picornavirales*, *Sobelivirales*, *Durnavirales*, *Tolivirales*, *Nodamuvirales*, *Wolframvirales*, *Ortervirales* and *Muvirales*) and families (*Picornaviridae*, *Marnaviridae*, *Dicistroviridae*, *Solemoviridae*, *Partitiviridae*, *Tombusviridae*, *Nodaviridae*, *Narnaviridae*, *Caulimoviridae* and *Qinviridae*), with most of them being sufficiently divergent from known viruses. The listed orders and families include a variety of RNA viruses with a wide range of hosts. Overall, the four examined pool samples were similar despite the large distances between sampling stations (Figure 1). This can be explained by the active mixing processes of the Lake Baikal waters, including those due to the currents and winds [45], as well as the long-term circulation of a wide variety of viruses in the lake’s ecosystem.

In the previous study [28], the authors used metatranscriptomic data to identify actively transcribing DNA viruses (those with an active process of RNA synthesis) in diverse invertebrate species; they managed to identify ssDNA virus species from the families *Parvoviridae*, *Circoviridae* and *Genomoviridae*, and dsDNA virus species from the families *Nudiviridae*, *Polyomaviridae* and *Herpesviridae*. In the Baikal sets, we identified viral sequences close to known DNA viruses of mollusks (for example, *Herpesviridae*); however, we failed to confirm that they belong to established viruses. We determined the similarity of almost all sequences identified as viral DNA to the assembled genomes of different gastropods (Appendix A) or to transposons. Eukaryotic genomes contain a great number (up to 85% of the genome in maize [46]) of transposable elements (TEs); many viruses can also integrate into the host genomes, leading to the exchange of genetic information. Examples of evolutionary interactions between TEs and viruses include ssRNA-RT viruses and *LTR* retrotransposons, dsDNA viruses (virophages and nucleocytoplasmic large DNA viruses, NCLDVs) and ‘*Polintons*’ transposons (or Maverick elements [47]), as well as the recently discovered ‘*Teratorn*’ transposon consisting of a ‘*piggyBac-like*’ transposon and the whole genome of herpesvirus [48]. In our study, the revealed similarity between the genes of invertebrates (mollusks) and RNA/DNA viruses most likely indicates a long history of coexistence and coevolution of the genomes of viruses and their hosts.

Only one scaffold was shown to represent a genome fragment of a ssDNA virus related to *Lake Sarah-associated circular molecule 10* (30.2% similarity to the *Rep* protein), recovered from a New Zealand freshwater Lake Sarah [13]; the closest BLAST hit using the nr database was CRESS virus sp. *ctWOo3* (QGH73619.1) from USA freshwater springs (58.6% similarity) (Figure 5). As is known, a variety of CRESS DNA viruses were identified from different mollusks species in New Zealand waters [13,49].

The list of revealed RNA virotypes included *Biomphalaria virus 1* (BV1) and *Biomphalaria virus 3* (BV3) (unclassified *Picornavirales*), the complete genomes of which were revealed in the study of freshwater aquatic pulmonate gastropods of the family Planorbidae, *Biomphalaria glabrata* and *Biomphalaria pfeifferi* [11]. The almost complete assembly of genomes and similarity of the *B. baicalensis* viruses BM_632 and BM_458 with BV1 and BV3, as well as with other related viruses from mollusks (Table 2, Figure 5), indicated, on the one hand, their non-random presence in samples of Baikal gastropods, and on the other hand, the wide distribution of viruses such as BV1 and BV3.

As is known, the diversity of picorna-like viruses and the number of their genomes have been actively expanded in the last decade through the analysis of viromes and transcriptomes of invertebrates (including mollusks) [14,15,17,24,25,26,27,50,51]. We also managed to uncover a new range of previously unknown picornaviruses. Some of the identified viruses could be present in the gastropod samples due to bioaccumulation through passed water, food or associated parasites [52]; however, there is ample evidence in favor of that picorna-like viruses may be responsible for mass mortality events in several species of marine mollusks [10]. The first picorna-like virus infection of shellfish was reported in 1986, when histopathological and ultrastructural studies were carried out in the marine mussel *Myths edulis* from Denmark [53]. A number of other studies [10] have also reported virus-like particles and virus-associated lesions similar to those described in [53], for example, in the pearl oyster, *Pinctada margaritifera*, from French Polynesia [54], in the carpet-shell clam, *Ruditapes decussatus* [55] or in cockles, *Cerastoderma edule*, from Galicia (Spain) [56].

In general, the similarity of proteins of *B. baicalensis* viruses to known viruses from the NCBI database was low and in single cases exceeded 70%. However, a search for similar sequences using BLASTp showed 99.3% identity and 99% query coverage of BM_14418 scaffold with unstructured polyprotein of *Tiger puffer nervous necrosis virus* (TPNNV, family *Nodaviridae*). *Nodaviridae* infect vertebrates or invertebrates and include three genera: *Alphanodavirus* (infects insects), *Betanodavirus* (fish) and *Gammanodavirus* (prawns). TPNNV, together with other representatives of the *Betanodavirus* genus, is known to infect over 60 different species of marine and freshwater fish, causing viral nervous necrosis, also known as viral encephalopathy and retinopathy. This disease was first described in the late 1980s; it causes fish mortality outbreaks and large economic losses to the marine aquaculture industry worldwide [57,58].

It was previously shown that invertebrate species, including mollusks, can serve as a reservoir for betanodaviruses [57]. In the phylogenetic analysis of RdRp domains, scaffold BM_14418 (close to *Tiger puffer nervous necrosis virus*) clustered with other betanodaviruses (MAGs) from bivalve mollusks (Figure 5). The bioaccumulation of another fish-pathogenic betanodavirus, *Red-spotted grouper nervous necrosis virus* (RGNNV), in hepatopancreatic tissue of manila clam *Ruditapes philippinarum*, and the ability to release infectious viral particles via fecal matter and filtered water was also shown [58]. Reservoirs of ‘nervous necrosis viruses’ were reported in different mollusks (bivalves, gastropods and cephalopods) and other marine invertebrate species (crabs, shrimps, artemias and rotifers) in waters of South Korea, Japan and China, in the European Atlantic and Mediterranean waters [59]. Baikal mollusks may also act as reservoirs for betanodaviruses (and possibly for other nodaviruses) and influence their epidemiology. We also discovered the scaffolds (BM_14418, BM_37266 and BM_13383) related to the alphanodavirus *Nodamura virus* (Table 2) and other noda-like viruses from crustaceans, other invertebrates and vertebrates (Figure 5c), but the low percentage of similarity does not allow us to confidently determine the taxonomy and host for these novel viruses, as well as for most other viruses (genomes/genome fragments) found in Baikal mollusks. The studies [60,61] revealed that the alphanodavirus of shrimps (*Covert mortality nodavirus*, CMNV) could switch their hosts to fish and sea cucumber (Echinodermata) by cross-species transmission, providing evidence for the broader host range of CMNV. New data obtained from numerous metagenomic studies, including our study, expand the range of diversity of nodaviruses; therefore, it is considered necessary to revise the generally accepted classification of these viruses [62].

Phylogenetic reconstruction of the evolution of RNA viruses (including numerous MAGs) based on the *RdRp* gene (4617 *RdRps*), as the only universal gene among RNA viruses, revealed the relationships between viruses of a wide range of hosts and identified five major branches [37]. Despite the constant expansion of the general phylogenetic tree and the addition of new groups, the established five branches still remain [38]. In our analysis, the tree included two of those branches: 2 and 3. Branch 2 (‘picornavirus-like supergroup’) includes eukaryotic RNA viruses from the orders *Picornavirales*, *Nidovirales*, the families *Solemoviridae*, *Caliciviridae*, *Potyviridae*, *Astroviridae*, a clade of dsRNA viruses, in particular, partitiviruses and picobirnaviruses; and some smaller groups of RNA and dsRNA viruses [37]. Of those listed, our tree contained picorna-like, solemo-like, and partiti-like viruses, similar to viruses from Baikal gastropods. Branch 3 consists of +RNA viruses, including a variety of tombusviruses and nodaviruses, the ‘Alphavirus supergroup’, ‘Flavivirus supergroup’ and several additional groups [37]. In our analysis, the tombus-like and noda-like MAGs and scaffolds were revealed in this branch.

*B. baicalensis* lives on rocky and mixed sandy-stony biotopes and is an omnivorous species that grazes on the substrate. Plant food (planktonic and benthic diatoms) and animal detritus (remains of worms, crustaceans and insect larvae) were found in the food bolus of these snails. It was shown that gastropods *B. baicalensis* pass sediments through their intestines and are able to capture pieces of tissue from dead fish [29]. In addition, Baikal gastropods widely use bacteria that colonize the surface of soil and stones as food [31]. All this contributes to the transmission and bioaccumulation (intake of viral particles by mollusk with water or food, their concentration and long-term retention in the intestine or other parts of the body [63]) of various viruses that infect a wide range of benthic and planktonic inhabitants of Lake Baikal. However, it is possible that some of the viruses identified in the metatranscriptomic data infect mollusks (to varying degrees, and perhaps asymptomatically), for example, sobemo-like or tombus-like viruses. Despite the undoubted successes of genomic research, the main problem remains determining the hosts for newly identified viruses. The search for hosts is complicated by evolutionary changes and rearrangements of viral genomes, transitions to new hosts, exchange of genetic material, etc. All this is becoming a new challenge for genetic and bioinformatics research.

As is known, mollusks can be a reservoir of vertebrate viruses, including humans, for example, noroviruses and hepatitis A virus [64] and others [6,7]. In our study, there were no human viruses in datasets, even in samples from the highly recreational area, Listvennichny Bay (sampling station #1). Only distant similarities were found between Baikal scaffolds and picornaviruses of carnivores (*Canine kobuvirus*) and birds (*Blackbird arilivirus*) (38.3% and 35.9%, respectively). It was previously shown that the canine distemper virus (*Paramyxoviridae*, *Morbillivirus*), which caused an epizootic outbreak in the Baikal seal (*Pusa sibirica*) population in 1987/88 [65,66], can accumulate in gastropods of the families *Baicaliidae* and *Lymnaeidae* without loss of infectivity [67,68]. The Ushkany Islands (sampling station #2) are a gathering place for Baikal seals, but no morbillivirus-like sequences were found in our metatranscriptomic data. It is possible that during the inter-epizootic period (the disease is not currently registered) concentration of viruses in lake water and mollusks (if available) is very low and there is not enough read depth to detect them.

As shown, the majority of viruses similar to the Baikal ones were uncharacterized viruses (MAGs) obtained from different environmental samples. A large number of viruses most closely related to those from Lake Baikal were discovered from freshwater organisms and sources, even from very distant geographical locations. We previously identified the same trend when analyzing RNA viruses from Baikal sponges [69]. This indicates that the viral diversity in natural aquatic ecosystems has not been yet fully explored, and that the habitat not only influences the formation of unique flora and fauna [70] but also the viral communities in the lake ecosystem.

## 4. Materials and Methods

### 4.1. Sampling Sites and Total RNA Sequencing

Mollusks of the genus *Benedictia* (*Benedictia baicalensis*) were taken at two stations of Southern Baikal: in Listvennichny Bay (near the village of Listvyanka) and near the village of Bolshie Koty, and near the Ushkany Islands in Northern Baikal (Table 1). Listvennichny Bay and Bolshiye Koty village are areas with different levels of anthropogenic load. On the coast of Listvennichny Bay there is one of the most accessible, popular and favorite places for tourists, the village Listvyanka. In summer, activity and the number of tourists increase, so we took samples in this place twice—in the cold (before summer) and warm (after summer) seasons on Lake Baikal. The Ushkany Islands are a gathering place for the Baikal seal; this station was interesting in terms of searching for viruses of these mammals in mollusks.

Sampling was carried out by divers. The mollusks were kept in the cold Baikal water for at least 24 h. Then the samples were washed in sterile Baikal water and frozen in liquid nitrogen for transportation and storage.

Total RNA was isolated from samples *B. baicalensis* (pools of five) using TRI Reagent (Molecular Research Center, Cincinnati, OH, USA) and a Direct-zol RNA MiniPrep kit (Zymo Research, Irvine, CA, USA). The quality and concentration of nucleic acids were assessed using gel electrophoresis (in 1.5% agarose), a Qubit 4.0 fluorimeter (Thermo Scientific, Waltham, MA, USA) and a Nanodrop 1000 spectrophotometer (Thermo Scientific, Waltham, MA, USA). Libraries were prepared using the MGI Easy RNA Library Prep Kit (MGI Tech Co., Ltd., Wuhan, China) in accordance with the manufacturer’s protocols. For each sample, two independent cDNA libraries were prepared and sequenced. The cDNA libraries were sequenced from both ends on a DNBSEQ^TM^ (DNA nanoball) platform using a DNBSEQ-400 NGS sequencer (MGI Tech Co., Ltd., Wuhan, China) and a DNBSEQ-G400RS High-throughput Sequencing Kit (MGI Tech Co., Ltd., Wuhan, China). The length of the sequenced fragments was 150 bases. Each cDNA library contained from 25 to 34 million read pairs.

### 4.2. Primary Processing of Reads

The quality visualization of datasets (paired reads) was carried out using the FASTQC v.0.11.9 program. Trimming of reads by quality was carried out with the Trimmomatic v.0.32 program (MAXINFO:40:0.05 AVGQUAL:15 MINLEN:90) [71]. After trimming, two replicates of sequenced libraries from the same sample were combined; the datasets finally contained between 63 and 67 million read pairs.

### 4.3. Taxonomic Analysis of Original Genome Reads

Taxonomic classification was carried out with the free software Kaiju v.1.9.0 [32] (https://github.com/bioinformatics-centre/kaiju (accessed on 25 August 2023)) using the NCBI nr (non-redundant) protein database. To increase the sensitivity of the Kaiju software, the parameter length was specified as 6, and the mismatch parameters were specified as 30. To reduce false positive results, the E-value was chosen as ≤0.00001 and bit score ≥ 50. The results of the Kaiju analysis were grouped in a table (Kaiju taxon count table), where the elements contained information about the representation of each taxon in the number of reads per sample. Based on the taxonomic identifiers, information on RNA viruses was extracted from the Kaiju taxon count table in the form of a table of virotype representation. For further analysis, relative values were used from this table (percentage of sample reads per RNA virotype).

### 4.4. Assembly of Metatranscriptomic Reads

All Baikal mollusk metatranscriptomic data (Table 1) were aggregated into one array for *de novo* cross-assembly as recommended in other viral community studies [72]. For cross-assembly, the MEGAHIT v.1.2.9 NGS assembler optimized for metagenomics was used with default options [34]. MEGAHIT avoids the formation of chimeric scaffolds from fragments of genomes of different species. The scaffolds with length ≥ 500 nucleotides were used for further analysis.

### 4.5. Viral Scaffolds Detection

We identified the viral scaffolds and open reading frames (ORFs) within them using the VirSorter2 v.2.2.4 tool [35]. VirSorter2 implements several approaches for identifying viral scaffolds in metagenomic assemblies, allowing increasing the accuracy of the analysis. The results of virus identification were checked by the CheckV program [73], according to the instructions (https://bitbucket.org/berkeleylab/checkv/src/master/, accessed on 23 December 2022).

### 4.6. Filtering out False Positive Viral Scaffolds

All scaffolds identified as viral by VirSorter2 were compared with chromosomal genome and proteomic data of mollusks from the NCBI database (Appendix A). To compare genomic data, the BLASTn (standard nucleotide BLAST) algorithm was used (word size 15, gapopen 2, gapextend 1, reward 1, penalty 1). A scaffold identified by VirSorter2 was considered a fragment of the mollusk genome if there was a hit for it with an E-value ≤ 0.00001, covering at least 30% of its length. To compare mollusk proteomes with viral scaffolds, the DIAMOND v2.0.14.152 tool [74] with a ”more-sensitive“ option was used. A scaffold identified by VirSorter2 was considered a fragment of the mollusk proteome if there was a hit for it with an E-value ≤ 0.00001, covering at least 300 nucleotides of its length. All viral scaffolds with selected thresholds of similarity to the genomes and proteomes of mollusks were removed from the analysis.

The viral scaffolds remaining after analysis of similarities with genomes and proteomes of mollusks were checked for mobile genetic elements, which can be mistakenly recognized as elements of viral genomes. To search for mobile elements, the DFAMSCAN program was used [36] with default parameters. For our analysis, HMM profiles of the following organisms were used: *Anopheles Coluzzii*, *Caenorhabditis Elegans*, *Danio Rerio*, *Drosophila Melanogaster*, *Hhalyomorpha Halys*, *Heliconius Erato Demophoon*, *Heliconius Melpomene*. A scaffold was considered a mobile element and retired from the analysis with an E-value ≤ 0.00001.

### 4.7. Taxonomic Assignment of Viral Scaffolds

Taxonomic identification for the viral scaffolds was carried out by comparisons of predicted viral proteins in scaffolds with the NCBI RefSeq [75] complete viral proteome database. The comparison was carried out by the DIAMOND [74] algorithm with ”more-sensitive“ option, E-value ≤ 0.00001 and bit score ≥ 50. For each protein in the scaffold, the best match in terms of the bit score value was selected. If a single scaffold had multiple proteins that matched different taxa (NCBI RefSeq ID), the one with the highest number of matching proteins was selected as the most closely related viral taxon from the NCBI RefSeq database (virotype). If the proteins were not repeated in the match list, the level of similarity of the matched proteins was taken into account, and the NCBI RefSeq taxon (ID) with the highest percentage of protein similarity was selected as the virotype.

The Bowtie2 v.1.3.1 (with default options) [76] and SAMtools v. 1.7 [77] results were used to determine the number of reads mapped on each predicted viral scaffold from each sample. Counts of the predicted viral proteins (ORFs) in samples were defined as the number of reads mapped on a scaffold containing a given protein. Consequently, the count table of viral scaffold representation in the analyzed samples was constructed. TPM (transcripts per million) normalization, recommended for metagenomics, was used to normalize the count table for scaffold length and number of reads per sample [78]. The TPM value for scaffold per samples was transformed (sum of TPM value for different scaffolds with the same virotype) into a table of TPM values per virotype in samples.

### 4.8. Functional Assignment of Viral Communities

The predicted viral proteins (ORFs) were matched with functional motifs in the Pfam database using PfamScan v.1.5 software [79] and the Conserved Domain Database [80] with default options. The ORFs and functional domains detected in the scaffolds were visualized to scale (Figure 4); reference genomes of closely related viruses (virotypes) from the NCBI RefSeq database (accession numbers NC_032804, NC_032589, NC_032840, NC_013460 and NC_013461) were processed in a similar way (Figure 4).

### 4.9. Phylogenetic Analysis of RNA-Dependent RNA Polymerase Genes

The search for data for phylogenetic analysis of *RdRp* genes was carried out in two ways. First of all, viral scaffold RdRp ORFs were used for BLASTp (protein—protein BLAST) to search for similar sequences in the NCBI nr database with default parameters. For each scaffold, the first five BLAST hits were selected according to E-value ≤ 0.00001 and alignment similarity ≥ 35%, with subsequent deleting of duplicates. The choice of the top five hits was driven by the intention to identify the closest known viral sequences from the NCBI nr database. RdRp ORFs were then compared to the viral proteomes of the NCBI RefSeq database (downloaded from https://ftp.ncbi.nlm.nih.gov/refseq/release/viral/ (accessed on 10 November 2023)) using the DIAMOND [67] algorithm (“more-sensitive” option). For RdRp ORFs, the DIAMOND hits were selected according to an e-value ≤ 0.00001, similarity ≥ 35%, alignment covering ≥ 70% of the ORF length and those defined to the species according to ICTV taxonomy (International Committee on Taxonomy of Viruses; https://ictv.global/ (accessed on 15 November 2023)). Thus, our analysis included NCBI nr BLAST-hits and known, well-studied viruses (with a full range of biological characteristics according to ICTV) from the RefSeq database with a significant degree of amino acid sequence similarity to the viruses we identified.

The final dataset comprised 245 amino acid sequences (44 scaffolds, 181 NCBI nr hits and 20 NCBI RefSeq hits). Aligning of sequences was carried out in MAFFT v.7.490 [81]. Phylogenetic reconstruction was performed with IQTREE v.1.6.12 [82]. The best-fitting amino acid substitution model (LG + G_4_ + Inv + F) was selected based on the BIC criterion values calculated by ModelFinder [83] implemented in IQTREE. Visualization was performed with FigTree (https://github.com/rambaut/figtree (accessed on 20 November 2023)).

### 4.10. Statistical Analysis of Taxonomic Diversity

The Kaiju proportion of reads for virotypes and TPM per samples of the Baikal mollusks were standardized into the ranges from 0 to 1 using the decostand function of the vegan package for R [84]. Standardized values of the Kaiju proportions of reads for virotypes and virotypes TPM per sample were visualized using canonical correspondence analysis (CCA). Gradient vectors of the viral family composition and scaffolds TPM per sample composition were fitted to a CCA scatter plot. Biodiversity analysis and CCA ordination were carried out using the vegan package for R [84], according to the instructions.

The Kaiju proportion of reads for the first 30 dominant virotypes and virotypes TPM per samples of the Baikal mollusks were visualized with a heat map using the gplots package [85] for R with columns (samples) clustering, and were grouped in order of similarity (Euclidean distance with the complete-link clustering method).

## 5. Conclusions

Newly identified viruses from Baikal mollusks have expanded the unique diversity of viruses previously discovered in the waters of Lake Baikal [86,87] and in the endemic Baikal sponges [69,72]. The viral diversity in Baikal endemic gastropods *B. baicalensis* included the unknown representatives of different orders and families and of two main phylogenetic branches [37]. Among the viral genomes (or scaffolds) identified were picorna-like viruses closely related to the previously described freshwater shellfish viruses *Biomphalaria virus 1* and *Biomphalaria virus 3*. Some other viruses were related to MAGs from other mollusks, and it is also possible that they are also pathogenic for gastropods. There is a theory of “host sharing and switching events” across different trophic levels during virus evolution [24], our and other metagenomic and metatranscriptomic studies were consistent with it. However, more in-depth studies are required to establish the actual host range for known and novel viruses. Mollusks can bioaccumulate viral particles passed through them with food, parasites or substrate and serve as a reservoir for viruses of different organisms, including vertebrates. All this complicates the discovery of the true virome of mollusks and presents new challenges for future research.

## Figures and Tables

**Figure 1 ijms-24-17022-f001:**
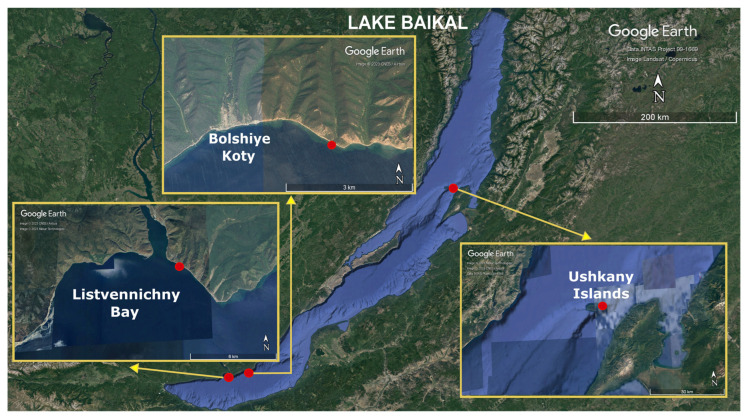
The map of Lake Baikal and sampling sites (indicated by red spots).

**Figure 2 ijms-24-17022-f002:**
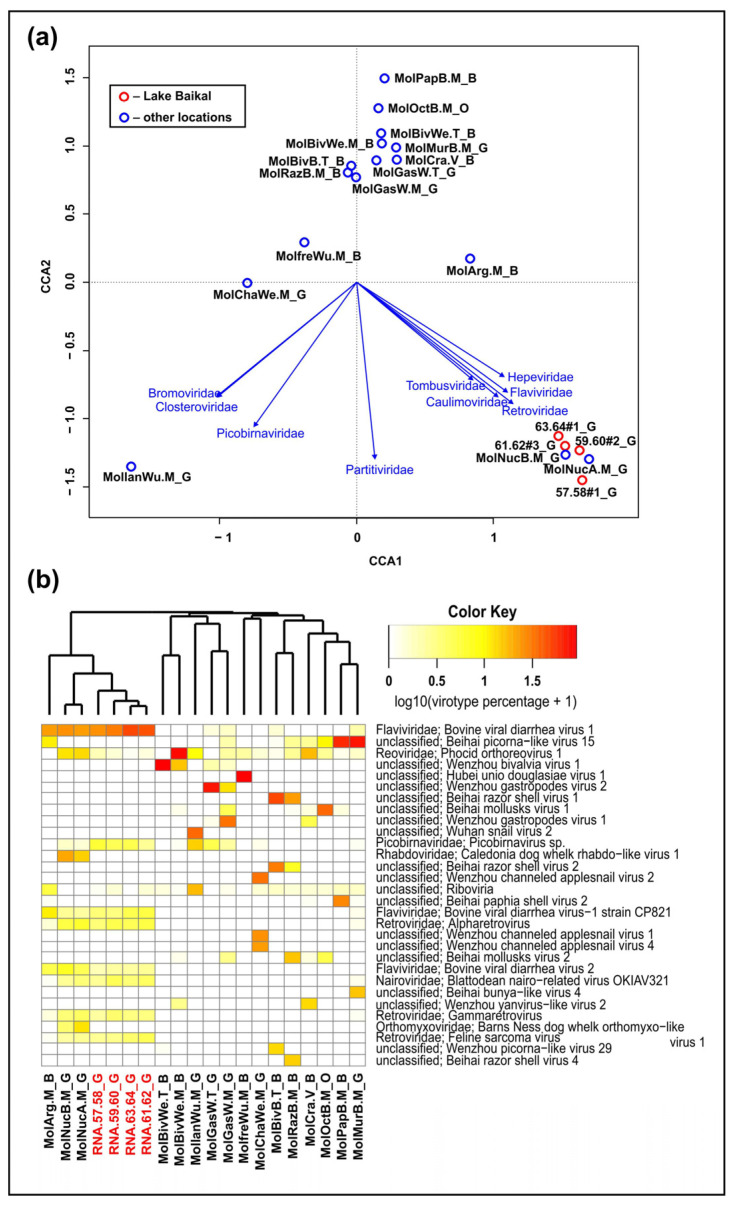
Comparative analysis of Baikal and other datasets from different mollusk species: (**a**) CCA analysis (canonical correspondence analysis) with reliable vectors of viral RNA families; (**b**) Heat maps of the 30 most abundant RNA virotypes based on the number of reads. The samples of Baikal mollusks are highlighted in red. Abbreviations (at the end of sample names): G—gastropods, B—bivalves, O—octopods.

**Figure 3 ijms-24-17022-f003:**
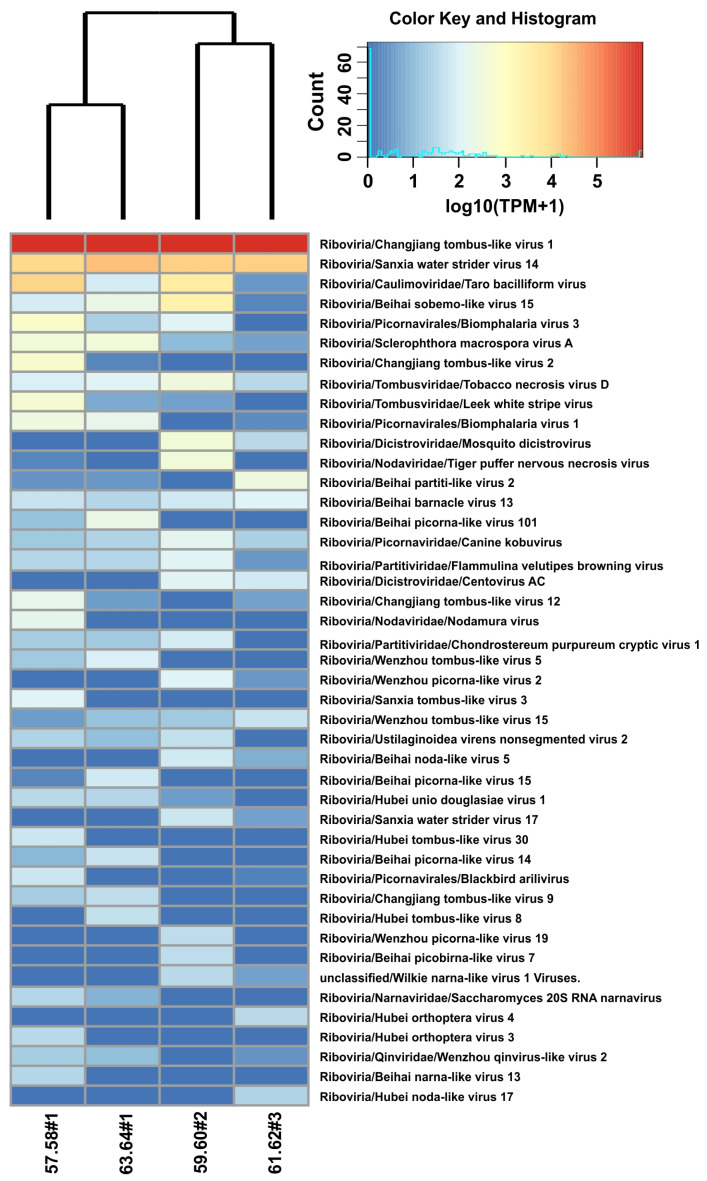
Heat map demonstrating the diversity of virotypes and their representation in the samples of the *B. baicalensis*. Heat map cells show normalized TPM (transcripts per million) values of the representation of virotypes in different mollusk samples. For convenience, TPM values have been converted to a decimal logarithm scale (log10(TPM + 1)). The color key contains a histogram of the occurrence of cells on the map with the corresponding log10(TPM + 1) values.

**Figure 4 ijms-24-17022-f004:**
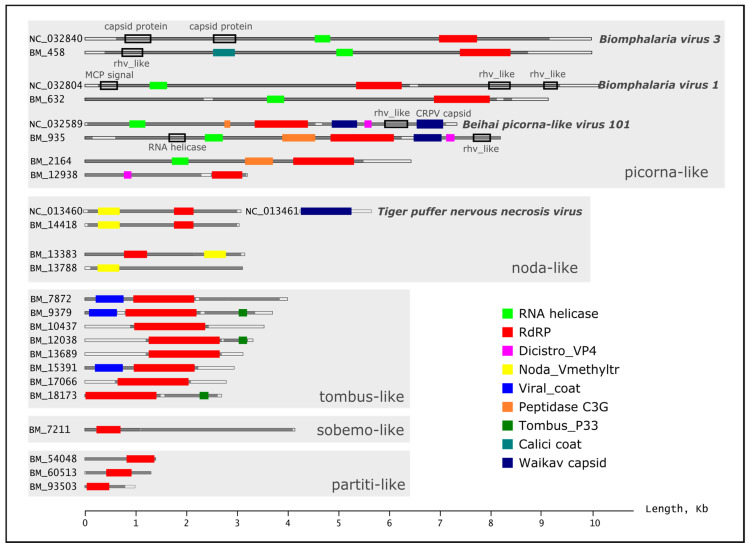
The longest viral RNA scaffolds from *B. baicalensis* samples, representatives of different groups of viruses. Almost complete genomes are shown along with those of virotypes (the closest relatives from the NCBI RefSeq database). Pfam domains are shown with color-filled boxes; additionally established CDD domains are shown with outlines, and their names are indicated.

**Figure 5 ijms-24-17022-f005:**
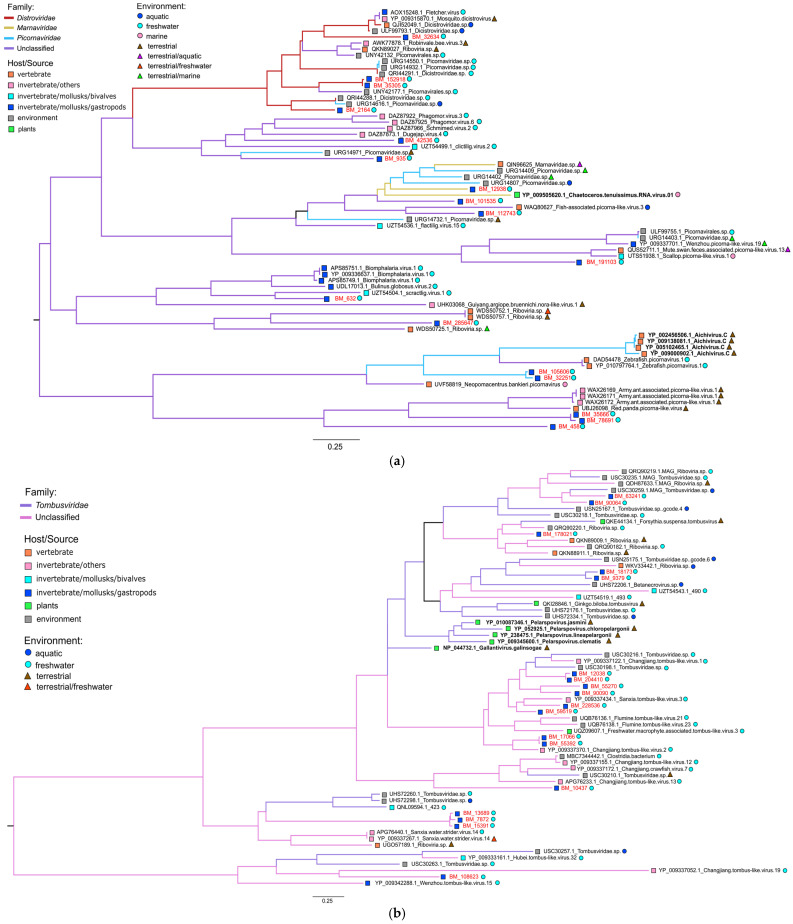
Phylogenetic trees based on the RdRp domain analysis (the RdRps of the Baikal viruses are highlighted in red, known ICTV viruses are highlighted in bold). Clusters correspond to the following orders: (**a**) *Picornavirales*, (**b**) *Tolivirales*, (**c**) *Nodamuvirales*, (**d**) *Durnavirales* and (**e**) *Sobelivirales*. Scale bars are expressed in the number of amino acid substitutions per site.

**Table 1 ijms-24-17022-t001:** Output data for samples of mollusks *Benedictia* sp. used for RNA sequencing.

Sample_ID	Sampling Site	Date	Depth, m	Latitude and Longitude
57.58#1	#1—Listvennichny Bay (Listvyanka)	31 October 2022	3–15	51°51′51.77″ N 104°50′37.80″ E
59.60#2	#2—Ushkany Islands	6 June 2021	2–9	53°51′05.76″ N 108°42′28.46″ E
61.62#3	#3—Bolshie Koty	31 May 2022	4–8	51°54′08.66″ N 105°06′13.04″ E
63.64#1	#1—Listvennichny Bay	10 June 2022	3–15	51°51′51.77″ N 104°50′37.80″ E

**Table 2 ijms-24-17022-t002:** Main characteristics of the longest (>2000 bp) scaffolds affiliated to RNA viruses: number and similarity of ORFs/proteins, closest relatives in the NCBI RefSeq/nr database, number of reads (TPM) per sample.

Scaffold ID_Length (nt)	Number of ORFs	Similarity of ORFs with RefSeq Proteins	BLAST Hit_RefSeq (Virotype)	Virotype Taxa (Known Order or Family)	Host/Source Lineage	TPM per Sample			
						57.58	59.60	61.62	63.64
BM_458/8746	1	24.8	*Biomphalaria virus 3*	*Picornavirales*	Mollusca; Gastropoda	550	0	0	3
BM_632/8432	3	29.6–32.2	*Biomphalaria virus 1*	*Picornavirales*	Mollusca; Gastropoda	250	0	1	183
BM_935/8225	3	25.5–29	*Beihai picorna-like virus 101*	unclassified	/Arthropoda; Decapoda	12	0	0	203
BM_2164/5832	2	33.2	*Centovirus AC*	*Dicistroviridae*	Insecta; Kerteszia	0	116	0	0
BM_7211/4124	2	28.1–39.3	*Beihai sobemo-like virus 15*	unclassified	/Arthropoda; Decapoda	71	2376	1	192
BM_7872/3857	4	30.6–49.2	*Sanxia water strider virus 14*	unclassified	/Insecta; Hemiptera	10,958	14,615	13,680	23,072
BM_9379/3384	3	38.3–45.3	*Tobacco necrosis virus D*	*Tombusviridae*	Viridiplantae; Magnoliopsida	91	323	31	112
BM_12038/3224	4	32.4–67.3	*Changjiang tombus-like virus 1*	unclassified	/Arthropoda; Decapoda	966,801	969,290	978,336	970,063
BM_12938/3213	2	42–49.8	*Beihai picorna-like virus 15*	unclassified	/Mollusca; Octopoda	1	0	0	64
BM_10437/3187	3	31.4–43.0	*Changjiang tombus-like virus 12*	unclassified	/Arthropoda; Decapoda	168	0	3	2
BM_13788/3115	1	33.9	*Sclerophthora macrospora virus A*	unclassified	Oomycota; Sclerophthora	299	9	0	334
BM_13383/3065	2	45.2–52.8	*Nodamura virus*	*Nodaviridae*	Insecta; Diptera	156	0	0	0
BM_14418/3051	1	99.3	*Tiger puffer nervous necrosis virus*	*Nodaviridae*	Vertebrata; Tetraodontiformes	1	329	0	0
BM_13689/2980	3	33.9–40.6	*Sanxia water strider virus 14*	unclassified	/Insecta; Hemiptera	1	9	386	1
BM_17066/2761	3	39.3–76.8	*Changjiang tombus-like virus 2*	unclassified	/Arthropoda; Decapoda	387	0	0	1
BM_15391/2722	3	39.3–46.8	*Sanxia water strider virus 14*	unclassified	/Insecta; Hemiptera	0	16	417	3
BM_18173/2610	2	47.3	*Leek white stripe virus*	*Tombusviridae*	Viridiplantae; Liliopsida	460	4	0	5
BM_23235/2360	1	23.7	*Hubei tombus-like virus 30*	unclassified	/Arthropoda; Arachnida	58	0	0	0

## Data Availability

Unprocessed datasets (raw reads) from samples of Baikal mollusks were submitted to the NCBI SRA database (BioProject PRJNA1029953, BioSamples SAMN37882679–SAMN37882682).

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
