# Peer review of "Viral Diversity in Samples of Freshwater Gastropods Benedictia baicalensis (Caenogastropoda: Benedictiidae) Revealed by Total RNA-Sequencing"

_ijms, 2023, doi:10.3390/ijms242317022_

Round 1

Reviewer 1 Report

Comments and Suggestions for Authors

This is an interesting, well written manuscript describing RNA viruses identified in a species of freshwater mollusc and comparing these with mollusc viruses identified in other studies. A number of novel RNA viruses are presented. The analysis is meticulous and well described and the results are presented clearly and accurately. The Materials and Methods section is very detailed, including for the bioinformatics analyses, which is very nice to see. None of the viruses identified are hugely divergent from known RNA viruses, however their presence in these hosts in this environment is of interest to others working in this field. The identification of betanodaviruses pathogenic to fish and the proposal that the molluscs act as a reservoir for these viruses is of particular interest. 

The approach used is fairly conservative and it is possible there are more viruses present in the samples than are discussed here, however this does not subtract from the results which are presented. 

My only major concern is with the phylogeny presented in Figure 4 - I think the viruses included are too divergent and dispersed for the phylogeny to be meaningful. My understanding is that the top five hits against each virus from NCBI protein have been included, but all that is really shown therefore is that these hits and the query sequences they were identified against cluster together in the tree. I would suggest making one tree for each of the five clusters and including representative known viruses (for example ICTV classified viruses from the same order) plus the previously described mollusc viruses mentioned in the introduction, where fall into these orders.  This would give much more context to show how the newly discovered viruses compare to known viruses. The paragraphs between lines 224 and 260, which are currently a little broad, could then be rewritten to more specifically put these new viruses into context. 

Another minor question, retroviruses are mentioned in Figure 2 but then are not discussed later in the paper - I’d like to know at which point these were filtered out. 

I have minor suggestions by line as follows:

12: Specify mollusc viruses

38: Reference needed for the 10E9 virus particles per millilitre value

107: It’s not very clear how the numbers here correspond to the information given in table S2 - in the table it seems like a large majority of reads are unclassified? The percentages given in the text seem unlikely given that this would mean almost zero reads were unclassified?

196: I’m not sure which 11 virotypes are being referred to here.

359-360: I’m not sure I understand the distinction between tombus-like viruses and true viruses here

367: I don’t agree that the smallest and simplest eukaryotic viruses are in the tombus-noda supergroup, the mitoviruses and narnaviruses are usually considered to be the simplest RNA viruses, infect plants and fungi (see e.g. doi 10.1016/B978-0-12-394315-6.00006-4) and fall into the Lenarviricota or branch 1 group.

442: independent rather than independently

507: How were Bowtie2 and Samtools used - which algorithms and settings?

552: This undersells the results slightly - only the Picorna-like sequences resemble well characterised mollusc viruses but many of the others are possible or likely mollusc viruses.

Materials and Methods - I would ideally like to see version numbers for all the software cited.

Reviewer 2 Report

Comments and Suggestions for Authors

The manuscript titled "Viral Diversity in Samples of Freshwater Gastropods Benedictia baicalensis (Caenogastropoda: Benedictiidae) Revealed by Total RNA-Sequencing" presents a well-structured study. The introduction establishes a solid context, the materials and methods section is detailed, and the results are informative. The discussion offers valuable insights, particularly concerning ecological and evolutionary relationships. The conclusions effectively summarize the research's significance and its implications for future studies.

Overall, the manuscript is well-written and accessible to a broad scientific audience. However, it's crucial for the authors to provide accession numbers for the raw data, which is currently missing. I recommend this manuscript for publication following major revisions. Below, I have provided several specific comments and suggestions for improvement.

Abstract:

The abstract should be more focused and structured for better readability.

The first sentence contains a grammatical error: "species" should be plural to match "shellfish." It should be "species that cause damage to food enterprises" or rephrased for better clarity.

The second sentence could be more concise and clear. It discusses the shift in focus from studying viruses in economically important species to studying viruses in natural populations of mollusks and other invertebrates. It would be beneficial to specify the relevance of this shift.

In the third sentence, the phrase "storehouse of natural viral diversity" is somewhat vague. It would be helpful to explain what is meant by "storehouse" and why invertebrates are important in this context.

In the fourth sentence, it would be beneficial to provide a concise overview of the research methodology used, such as Total RNA-Sequencing, to give readers an idea of how viral diversity was assessed.

The abstract mentions "mollusks collected at various stations of Lake Baikal," which is good information. However, it would be more informative if it briefly mentioned the sample size or range, which can give an idea of the scale of the study.

In sentence six, the specific virus "Tiger puffer nervous necrosis virus" is mentioned, but it's unclear whether it's directly relevant to the study's focus on gastropods. If relevant, explain the significance; if not, consider omitting this detail.

In the seventh sentence, the mention of "picorna-like viruses" and their potential pathogenicity for Baikal gastropods is crucial. However, it should be briefly elaborated to provide more context.

The final sentence discusses the possibility of mollusks serving as a reservoir for unidentified viruses. While intriguing, it could be stated more clearly to indicate its significance to the research.

Keywords:

The selected keywords are relevant to the study, but it might be helpful to include more specific terms related to the research, such as "metatranscriptomic analysis," "Lake Baikal," and "RNA-containing viruses."

Introduction

Line 31. You should consider formatting the list of mollusk classes in a bullet or numbered list for better readability.

Line 33. The phrase "and freshwater (7,000)" should be formatted consistently with the other numbers, for example, "and freshwater (7,000)."

Line 35. It would be helpful to briefly mention the importance of mollusks and their immunity in the context of the study.

Line 41. The phrase "Farley С.А. in review [5]" is unclear. It should be clarified whether it refers to a specific publication or research.

Line 46. Instead of listing families of viruses, it might be more effective to briefly explain that various viral families were identified in mollusks, including DNA and RNA viruses, to maintain the flow of the text.

Line 57. The importance of the three studied viral families to the production of bivalve mollusks should be elaborated.

Line 61. The mention of the 220 species should be clarified by stating that it includes mollusks and specifying the type of animals.

Line 65. The significance of "recombination, lateral gene transfer, and complex genomic rearrangements" in viruses and their hosts should be briefly explained.

Line 70. It would be beneficial to explain why the diversity of marine invertebrate RNA viruses is significant in the context of the study.

Line 71. Instead of "Metagenomic (shotgun sequencing) analysis is also used," it would be clearer to say something like "Metagenomic analysis, including shotgun sequencing, is employed..."

Line 83. Clarify why metatranscriptomic analysis was chosen for the study and how it relates to identifying viruses.

Line 85. The number of gastropod species in Lake Baikal is mentioned. Consider mentioning the importance of studying these specific gastropods.

Line 87. Explain how gastropods play a significant role in the biological processes of the lake.

Line 88. Provide a brief overview of Benedictia baicalensis and its role in the ecosystem.

Line 90. Specify the findings of the study and their significance to the research.

Line 92. Clarify what is meant by "distant similarities."

Line 94. Explain the importance of the noted relationship between Baikal viruses and those from other freshwater ecosystems.

Results

Line 100: It would be helpful to mention what "Baikal samples" refer to, providing a brief explanation.

Line 102: Specify what "Kaiju software" is and provide a brief overview of how it was used.

Line 107: It would be more informative to state the range of percentages (e.g., 1.3-1.7% and 1.3-2.0%) for eukaryotic, bacterial, and microeukaryotic reads, respectively.

Line 110: Clarify why only RNA virus reads were considered for further analysis. The reasons for this choice should be briefly explained.

Line 112: Explain why the samples of B. baicalensis were distant from the main part of the datasets and grouped with samples of marine gastropods. Elaborate on the significance of this finding.

Line 116: Provide more context for the distinctive sample, MollanWu.M, and its relevance to the study's objectives.

Line 139: The text starting from "as well as virus-similar sequences..." seems incomplete and lacks context. It should be revised to provide a clear transition to the next section.

Line 142: Mention the significance of using the MEGAHIT program and the VirSorter2 program for the analysis of assembled scaffolds.

Line 151: Explain the importance of retaining only scaffolds with sequences belonging to eukaryotic viruses for further analysis.

Line 155: Clarify why certain sequences identified as viral were excluded from further analysis.

Line 157: Specify the importance of analyzing the 44 identified RNA virotypes in the context of the study.

Line 159: It's important to elaborate on the significance of identifying these virotypes and the families to which they belong, especially in the context of the study.

Line 187: Provide a brief explanation of what the heat map demonstrates and why it is significant.

Line 204: The text should flow more smoothly into the mention of Figure 4, explaining its significance and what the figure shows.

Line 217: Mention the significance of selecting the top 5 BLASTp hits from the NCBI nr database for each RdRp domain.

Line 224: Clarify the implications of the even distribution of RdRp sequences from Baikal gastropods throughout the tree.

Line 245: Elaborate on the significance of the significant distances between the RdRp genes of Baikal viruses and those from other organisms and biotopes.

Line 247: Explain the relevance of the phylogenetic reconstruction of the evolution of RNA viruses based on the RdRp gene for the study's objectives.

Discussion

Line 263: "total RNA isolated" - The term "total RNA isolated" should be written as "total RNA isolated."

Line 267: "from different viral orders (Picornavirales, Sobelivirales, Durnavirales, Tolivirales, Noda-" - The text is cut off abruptly. Please complete this sentence to provide more information about these viral orders.

Line 274: "actively transcribing DNA viruses" - The phrase "actively transcribing DNA viruses" could be unclear to some readers. It might be helpful to briefly explain what is meant by "actively transcribing."

Line 282: "Eukaryotic genomes contain a great" - The sentence seems incomplete. It should be completed to convey the intended meaning.

Line 316: "mass mortality events in several species of marine mollusks" - It would be beneficial to provide some examples or references to support this statement.

Line 361: "organisms distant from plants" - It would be helpful to explain the significance of viruses related to plants being found in organisms distant from plants.

Line 374: "mammals, birds, and reptiles" - It might be beneficial to clarify that the phrase refers to parasites infecting these animals.

Line 385: "mollusks or other aquatic or semi-aquatic inhabitants" - The text could be enhanced by specifying some examples of these semi-aquatic inhabitants.

Line 398: "transmission and bioaccumulation of various viruses" - It would be helpful to explain the mechanism of transmission and bioaccumulation in more detail.

Line 407: "mollusks can be a reservoir of vertebrate viruses, including humans" - This statement should be supported with references or examples.

Line 415: "the titer of viruses is very low" - It would be helpful to clarify what "titer of viruses" refers to in this context for readers who may not be familiar with the term.

Line 419: "reads depth" - The term "reads depth" should be corrected to "read depth."

Line 421: "MAGs" - It would be beneficial to expand the acronym "MAGs" (Metagenome-Assembled Genomes) for clarity.

Materials and methods

Line 430: The genus "Benedictia" is mentioned in the text, but the term "Benedictia sp." is used. It's important to maintain consistency in nomenclature. Either use "Benedictia" or specify the species within this genus if it is essential.

Line 443: "DNB" should be spelled out as "DNA" in the context of DNA sequencing.

Line 459: "taxa" is the plural form of "taxon," so it should be "the representation of taxa."

Line 468: Mention the source of the "MEGAHIT NGS assembler" and provide a reference if necessary.

Line 485: "e-value" should be "E-value" for consistency.

Line 487: The abbreviation "DIAMOND" should be defined before use.

Line 488: "en e-value" should be corrected to "an E-value."

Line 514: "virotype" should be defined or explained for clarity.

Line 523: "BLASTp" should be defined before use.

Line 525: "BLAST" is an acronym and should be capitalized.

Line 533: It would be helpful to define what CCA stands for.

Line 546: "replenished" could be replaced with "expanded."

Line 556: It's suggested to clarify the term "MAGs" for readers who might not be familiar with this abbreviation.

Comments on the Quality of English Language

Minor editing of English language required.

Reviewer 3 Report

Comments and Suggestions for Authors

Butina et al. are interested in the virome in mollusks of Lake Baikal in Russia. Using the total RNA-sequencing method and metatranscriptomic analysis, they identified a plethora of genetically distant RNA viruses with known viruses in other host samples from freshwater. Most importantly, the authors found that some picorna-like viruses may be pathogenic for Baikal gastropods. Overall, this study is informative and well-designed. However, some points are not clearly clarified. This reviewer has the following comments that the authors could consider.

1.     Lines 84-86: as indicated by the authors, there are around 150 species of gastropods in Lake Baikal; why do the authors merely focus on the species Benedictia baicalensis? Is this species a unique gastropod resident in the Lake Baikal?

2.     Did the authors compare the virome in the three locations in Lake Baikal? Is there any difference? Why were the samples collected twice in Listvennichny Bay?

3.     Table 1: please specify the numbers in the “Sample” column. What do these numbers mean?

4.     Figure 1: this reviewer is confused by the white arrows, especially the Bolshiye Koty. Are the red spots supposed to be the sampling sites?

5.     Line 90: This sentence is not related to the Table 1 and Figure 1. Is there any explanation that most detected were positive-strand RNA viruses in B. baicalensis samples?

6.     Curiously, how did the authors conclude that the picorna-like viruses may be pathogenic for Baikal gastropods? The data presented in the study cannot convince this reviewer.

7.     The Discussion is really too long. Please consider condensing. In fact, much content could be moved to the Results section.

Comments on the Quality of English Language

Minor editing of English language required.

Round 2

Reviewer 2 Report

Comments and Suggestions for Authors

The authors meticulously addressed the revisions suggested by the reviewer. I'm very pleased with the adjustments made and wholeheartedly recommend this manuscript for publication in its current form.

Reviewer 3 Report

Comments and Suggestions for Authors

I am satisfied with the modifications made by the authors and have no further critical comments.